# Enhanced Slime Mould Optimization with Deep-Learning-Based Resource Allocation in UAV-Enabled Wireless Networks

**DOI:** 10.3390/s23167083

**Published:** 2023-08-10

**Authors:** Reem Alkanhel, Ahsan Rafiq, Evgeny Mokrov, Abdukodir Khakimov, Mohammed Saleh Ali Muthanna, Ammar Muthanna

**Affiliations:** 1Department of Information Technology, College of Computer and Information Sciences, Princess Nourah bint Abdulrahman University, P.O. Box 84428, Riyadh 11671, Saudi Arabia; 2School of Automation, Chongqing University of Posts and Telecommunications, Chongqing 400065, China; asn_rafiq@hotmail.com; 3RUDN University, 6 Miklukho-Maklaya Street, 117198 Moscow, Russia; mokrov-ev@rudn.ru (E.M.); khakimov-aa@rudn.ru (A.K.); muthanna.asa@spbgut.ru (A.M.); 4Institute of Computer Technologies and Information Security, Southern Federal University, 347922 Taganrog, Russia; muthanna@sfedu.ru

**Keywords:** unmanned aerial vehicles, slime mould algorithm, resource allocation, deep learning, wireless networks

## Abstract

Unmanned aerial vehicle (UAV) networks offer a wide range of applications in an overload situation, broadcasting and advertising, public safety, disaster management, etc. Providing robust communication services to mobile users (MUs) is a challenging task because of the dynamic characteristics of MUs. Resource allocation, including subchannels, transmit power, and serving users, is a critical transmission problem; further, it is also crucial to improve the coverage and energy efficacy of UAV-assisted transmission networks. This paper presents an Enhanced Slime Mould Optimization with Deep-Learning-based Resource Allocation Approach (ESMOML-RAA) in UAV-enabled wireless networks. The presented ESMOML-RAA technique aims to efficiently accomplish computationally and energy-effective decisions. In addition, the ESMOML-RAA technique considers a UAV as a learning agent with the formation of a resource assignment decision as an action and designs a reward function with the intention of the minimization of the weighted resource consumption. For resource allocation, the presented ESMOML-RAA technique employs a highly parallelized long short-term memory (HP-LSTM) model with an ESMO algorithm as a hyperparameter optimizer. Using the ESMO algorithm helps properly tune the hyperparameters related to the HP-LSTM model. The performance validation of the ESMOML-RAA technique is tested using a series of simulations. This comparison study reports the enhanced performance of the ESMOML-RAA technique over other ML models.

## 1. Introduction

The rising demand for higher-quality wireless services drives upcoming wireless transmission systems to be responsible for widespread coverage and connectivity over any mobile device [1,2]. Also, the variety of network applications greatly demands energy consumption, network capability, and service latency for masses of mobile devices. For realizing the vision of limitless access to wireless information anytime and anywhere for everything, the recently developed unmanned aerial vehicle (UAV)-based flying platform is capable of breaking the limitation of the conventional network structure that drives us to rethink the advancement of transmission systems in the upcoming generation [3]. UAVs—in other words, drones—have gained special consideration due to their low-cost deployment, simplicity, and prominent flexibility. Due to their higher flying attitude, drone-based platforms have established effectual Line-of-Sight (LoS) connection with ground users (GUs), thereby reducing the power utilization for reliable connectivity [4]. Figure 1 demonstrates an overview of UAV-enabled WSNs.

Thus, the UAV-based flying mobile transmission technique gives energy- and cost-effective solutions with a limited territorial cellular structure for the GU. Formulating drone-assisted wireless communication systems has attracted more and more research interest [5]. The present study on the UAV-related wireless communication mechanism primarily focuses on resource optimization and drone placement, assuming that drones serve as aerial relays or aerial BSs to assist GUs [6]. The altitude of a drone can be enhanced for the trajectory model without or with the horizontal location related to various QoS requirements and considerations.

Although UAV-related communications have numerous merits in real time, certain technical issues should be solved to unlock the promising performance gains [5]. Initially, stringent power limitations were a bottleneck to effective drone communications. The energy storage of the onboard batteries of a drone was generally small because of the size and weight limitations of the drone [7]. Moreover, transmission and flight power consumption are based on the drone’s velocity and trajectory. Thus, energy-efficient drones have drawn critical research attention in the literature. Binding the above-mentioned advantages of drone-based networks faces several technical challenges in resource allocation models [8,9]. To be specific, UAV-based networks perform optimally if drones’ trajectories or positions are adequately planned, drones’ power transfer is adequately assigned, and the UAV-UE relationship is properly managed to handle the dynamic of channel state information (CSI) among UEs and UAVs [10]. Such joint designs frequently need a precise CSI prediction. But perfect CSI is only sometimes probable since drones can flexibly move in space, making CSI rapidly change location over time. Moreover, UAV-based networks still suffer similar difficulties to multicast communication networks and coordinated multipoint (CoMP) [11]. More realistic methods for efficiently advancing drone-assisted networks are of timely importance.

This paper presents an Enhanced Slime Mould Optimization with Deep-Learning-based Resource Allocation Approach (ESMOML-RAA) in a UAV-enabled wireless network. The ESMOML-RAA technique considers UAVs as learning agents with the formation of resource assignment decisions as actions and designs a reward function to minimize weighted resource consumption. For resource allocation, the presented ESMOML-RAA technique employs a highly parallelized long short-term memory (HP-LSTM) model with an ESMO algorithm as a hyperparameter optimizer. Using the ESMO algorithm helps properly tune the hyperparameters related to the HP-LSTM model. The performance validation of the ESMOML-RAA technique is tested using a series of simulations. In short, the key contributions are listed as follows:Developing a new ESMOML-RAA technique for the optimal allocation of resources in UAV-assisted wireless networks, which comprises the effective allocation of restricted resources like bandwidth, power, and computing resources to multiple UAVs to optimize the system performance.Employing HP-LSTM for resource allocation, where LSTM is a type of recurrent neural network that can effectively capture long-term dependencies and temporal patterns in sequential data, making it suitable for modelling the dynamic nature of UAV networks.Designing an ESMO algorithm by integrating the concept of elite oppositional-based learning (EOBL), which enhances the exploration and exploitation capabilities of the SMO algorithm. Hyperparameter tuning using the ESMO algorithm helps improve the HP-LSTM model’s performance.

## 2. Related Works

Nguyen et al. [12] presented reconfigurable intelligent surface (RIS)-supported drone networks that either benefit the drone’s agility or employ RIS reflection to improve the network performance. A deep reinforcement learning (DRL) system was presented to resolve the continuous optimization issue with time-varying channels in a centralized way. Luong et al. [13] examined a new technique for developing the deep Q-learning (DQL) technique to consider the hassles of inaccessible CSI to determine the location of drones, and invoked the variance of convex (DC)-related optimization technique to proficiently overcome drone transmit beamforming and drone–user association to specify the determined location of drones. In [14], research scholars considered the minimized sum power issue by cooperatively enhancing power control, RA, and user association in an MEC system that includes many drones. The author developed a centralized multiagent RL (MARL) technique since the issue was nonconvex. But, essential problems, namely privacy concerns and distributed frameworks, were ignored by the centralized method. The authors modelled a multiagent federated RL (MAFRL) technique in a semidistributed structure.

Cui et al. [15] examined the dynamic RA of many drone-based transmission networks to maximize long-term benefits. The authors developed the long-run RA issue as a stochastic game to maximize the anticipated rewards and to design the uncertainty and dynamics in surroundings, in which all drones will be learning agents and all RA solutions correspond to an activity engaged in by the drones. Then, as per its local observations utilizing learning, the authors developed a MARL structure where all agents find their optimal method. In [16], the drone as a flying BS was considered for an emergency transmission system, including 5G mMTC network slicing, to enhance the service quality. The drone-related mMTC makes a BS selection method to maximize the system’s energy efficiency. Afterward, utilizing the Markov decision process (MDP) theory, the system method can be minimized into the stochastic-optimization-related issue. The authors devised an approach to optimize energy efficiency to solve the RA problem, a Dueling-Deep-Q-Network (DDQN)-related method related to the RL method.

The authors in [17] proposed a drone-assisted distributed routing framework focusing on quality-of-service provision in IoT environments (D-IoT). The study in this paper focused on highly dynamic flying ad hoc network environments. This model was utilized to develop a distributed routing framework. A neuro-fuzzy interference system was applied to achieve reliable and efficient route selection. A quality-of-service provisioning framework for a UAF-assisted aerial ad hoc network environment (QSPU) was proposed in [18] to achieve reliable aerial communications. UAV-centric mobility models were utilized to develop a complete aerial routing framework, and it was proved that a number of service-oriented performance metrics in a UAV-assisted aerial ad hoc network environment achieved better performance. Furthermore, for a UAV-assisted aerial ad hoc network environment, a quality-of-service provisioning framework was proposed to focus on reliable aerial communication [19]. UAV-centric mobility models were utilized to develop aerial routing frameworks.

Li et al. [20] devised a novel DRL-related flight RA framework (DeFRA) method for reducing the overall loss of data packets in continual action spaces. The abovementioned method depends on deep deterministic policy gradient (DDPG), optimally controlled speeds, and instantaneous headings of the drone and chooses ground devices for collecting data. Additionally, for predicting network dynamics resulting from energy arrivals and time-varying airborne channels in ground devices, a state characterization layer using LSTM was formulated. In [21], the authors deployed a clustered multidrone for providing RA and computing task offloading services to IoT gadgets. The author developed a multiagent DRL (MADRL)-related method for minimizing the overall network computational cost while assuring QoS necessities for UEs or IoT gadgets in the IoT platform. To reduce long-term computation costs with regard to energy and delay, the authors developed the issue as a natural extension of the Markov decision process (MDP) considering a stochastic game.

## 3. The Proposed Model

In this study, we developed a new ESMOML-RAA technique for resource assignment in UAV-enabled wireless networks. Figure 2 shows the working process of the proposed model. The presented ESMOML-RAA technique attained energy-effective and computationally effective decisions proficiently. At the same time, the ESMOML-RAA technique considered the UAV as a learning agent with the formation of resource assignment decisions as actions and designed a reward function with the intention of minimizing weighted resource consumption.

### 3.1. System Model

Consider a multi-UAV MEC network, wherein N mobile users (MUs) are randomly distributed with M UAVs flying in a specific place. Every UAV assumes computation and communication abilities, which enables MUs to offload the task [22,23]. The study aims to minimize the computation and energy utilization of UAVs via resource allocation (RA) using the QoS constraint of MUs concerning latency. The sets of UAVs and MUs are correspondingly indicated by $M=\left\{1,2,\;\ldots ,\;M\right\}$ and N = $\left\{1,2,\;\ldots ,\;N\right\}$. Then, the network functions are considered during a time K that comprises $T^{\gamma }$ time intervals represented as $T=\left\{1,2,\ldots ,\;T^{\gamma }\right\}$, and the location of the MU is considered to be constant in time intervals, signified as $q_{i}\left(t\right)=\left[x_{i}\left(t\right),\;y_{i}\left(t\right)\right]$. Moreover, the UAV is considered to be moving at a constant altitude H with R radius coverage, and the coordinate of the *j*th UAV is denoted by $p_{j}=\left[X_{j}\left(t\right),\;Y_{j}\left(t\right)\right]$. Additionally, the *k*th tasks of *i*th MU are specified as $R_{i}^{k}=\left(S_{i}^{k},\;F_{i}^{k},\;D\right)$, whereby $S_{i}^{k},\;F_{j}^{k},\;D$ represents the input size, the necessary CPU circle, and the maximal tolerant time of $k^{th}$ task, respectively. It should be noted that D is similar to every task that characterizes the QoS constraint of latency-intensive tasks. Table 1 shows some of the notation used in the proposed approach.


**Communication model**


The channel gain from *j*th UAV to *i*th MUs was modelled as well as the representation of a multi-UAV network infrastructure, where every UAV independently makes an RA decision.
(1)$$h_{i}^{j}\left(t\right)=\beta_{0}d^{-2}\left(t\right)=\frac{\beta_{0}}{H^{2}+|\left|q_{i}\left(t\right)-p_{j}\left(t\right)\right|{|}^{2}},\;\;$$

In Equation (1), $\beta_{0}$ indicates the channel gain at the reference distance $d_{0}=1$ m. Once UAV j handles *k*th tasks from MU i, the data of task should be transferred from *j*th to *i*th, and the throughput is given as follows:(2)$$r_{i,k}^{j}\left(t\right)=B\log_{2}\left(1+\frac{a_{i,k}^{j}\left(t\right)Ph_{i}^{j}\left(t\right)}{\sigma^{2}+I_{i}^{u}\left(t\right)}\right),\;\;$$

In Equation (2), $a_{i,k}^{j}\left(t\right)\in \left[0,1\right]$ indicates the energy allocation indicator, B represents the overall bandwidth allocated to MU, and P indicates the maximal transmission power of the UAV. Now, the radio resource utilized by the smaller cells is assumed to overlap; hence, mutual interference takes place after the similar task is transferred to distinct UAV servers, using $I_{i}^{u}\left(t\right)=\sum_{u\epsilon M,\;u\;\neq j}p_{u}\left(t\right)h_{i}^{u}\left(t\right)$. $\sigma^{2}$ represents the background noise power. Thus, the communication time of *k*th tasks in *t*th interval is given by the following:(3)$$L_{i,k}^{j,s}\left(t\right)=\frac{S_{i}^{k}\left(t\right)}{r_{i,k}^{j}\left(t\right)},\;$$

In Equation (3), $S_{i}^{k}\left(t\right)$ represents the size of the implemented task in the *t*th time interval. The transmission power consumption of the *j*th UAVs in the *t*th time interval is formulated by the following:(4)$$p_{i_{;}k}^{j,s}\left(t\right)=a_{i_{;}k}^{j}\left(t\right)PL_{i_{;}k}^{j,s}\left(t\right).\;$$

Generally, the computational resource of each UAV is considered to be the same, indicated as C circles for every second. Therefore, the execution time of the *k*th task in the *j*th UAVs is formulated by
(5)$$L_{i,k}^{j,c}\left(t\right)=\frac{F_{i}^{k}\left(t\right)}{b_{i,k}^{j}\left(t\right)C},$$

In Equation (5), $b_{i,k}^{j}\left(t\right)\in \left[0,1\right]$ signifies the computational RA decision, and it can be formulated as
(6)$$p_{i,k}^{j,c}\left(t\right)=\gamma_{0}b_{i,k}^{j}\left(t\right)C.\;$$

In Equation (6), $\gamma_{0}$ refers to the constant associated with the hardware structure.

### 3.2. Problem Formulation

By mutually considering energy consumption and computation complexity, the study focuses on minimizing the resource consumption of UAVs using the constraints of the MUs’QoS with respect to latency:(7)$$\underset{a,b}{\min}\omega_{0}\overset{M}{\underset{j=1}{\sum }}\overset{T}{\underset{t=1}{\sum }}l_{i,j,t}p_{i,k}^{j,s}\left(t\right)+\left(1-\omega_{0}\right)\overset{M}{\underset{j=1}{\sum }}\overset{T}{\underset{t=1}{\sum }}l_{i,j,t}p_{i,k}^{j,c}\left(t\right),\;\forall i\in N,\forall k\in K\;$$
$$s.t.\;C1:\;a_{i,k}^{j}\left(t\right)\in \left[0,1\right]\;C2:\;b_{i,k}^{j}\left(t\right)\in \left[0,1\right]\;C3:\;t_{i,k}^{j,c}\left(t\right)+t_{i,k}^{j,s}\left(t\right)\leq D\forall i\in N,\forall k\in K\;\;\;\;\;\;\;\;\;\;\;\;\;\;\;\;\;\;\;\;\;\;\;\;C4:\;|\left|q_{i}\left(t\right)-p_{j}\left(t\right)\right|{|}^{2}\leq R^{2}\;\;\;\;\;\;\;\;\;\;\;\;\;\;\;\;\;C5:\;\overset{M}{\underset{i=1}{\sum }}l_{i,j,t}\leq 1\forall j\in M,\forall t\in T\;\;\;\;\;\;\;\;\;\;\;\;\;\;\;\;$$

From the expression, $l_{i,j,t}=\left\{0,1\right\}$ indicates either MU i offloads tasks to UAV j at time interval t. C1 and C2 show the constraint on the radio RA and computation RA, respectively. C3 shows that the overall processing time of task k must fulfil the maximal tolerant time D. Meanwhile, the radius coverage of the UAV denotes R, and C4 guarantees the designated MU and UAV in the transmission range. C5 represents the fact that every UAV could implement the individual task at a particular time interval.

### 3.3. Resource Allocation Using the HP-LSTM Model

For resource allocation, the presented ESMOML-RAA technique employed the HP-LSTM model. For enabling the LSTM to compute $0^{t}$ in parallel, the HPLSTM utilizes a bag-of-words representation $s^{t}$ of previous tokens for the computation of gates and HL [24]:(8)$$s^{t}=\overset{t-1}{\underset{k=1}{\sum }}i^{k}$$
whereas $s^{1}$ refers to the zero vector. The BoW representation $s^{t}$ is attained effectually using the cumulative sum function. Figure 3 showcases the structure of LSTM.

The proposed neural network consists of several layers as depicted. The first layer is a unit vector layer used to change the input from number to vector form as required by the long-short-term memory (LSTM) implementation. The next layer is a recurrent LSTM layer with a memory parameter. The LSTM layer consists of several elements according to input, output, and forget gates which use logistic sigmoid, while the memory gate uses the tanh activation function. A single LSTM layer with all the gates is illustrated in Figure 3, where X_t_ is the input, h_t_ is the output on iteration, and C_t_ is the state of the layer on iteration. Last is a sequence last layer that returns the last element of the sequence. After that, there is a linear layer with n inputs and m outputs. The last layer is a softmax layer that normalizes the outputs.

Afterward, it concatenates the input i and the equivalent layer standardized BoW representation $LN\left(s\right)$ for succeeding in computation:(9)$$v=iLN\left(s\right)$$

Here, layer normalization is introduced to prevent potential explosions due to accumulation in Equation (8) to stabilize the training process.

Then, it calculates the input gate, forget gate, and HL:(10)$$i_{g}=\sigma \left(LN\left(W_{i}v+b\right)\right)\;$$
(11)$$f_{g}=\sigma \left(LN\left(W_{f}v+b_{f}\right)\right)\;$$
(12)$$h=\alpha \left(LN\left(W_{h}v+b\right)\right)\;\;$$

Then, v is calculated on the order before the computation of these gates and HLs; Equations (10)–(12) are only needed to calculate the entire order, allowing the effective sequence-level parallelization of higher-cost linear transformation. But the BoWs context representation $s^{t}$ lacks a weighting process related to the preceding step output $o^{t-1}$ of the novel LSTM; therefore, utilizing a two-layer feed-forward network for HL computation was also attempted to alleviate potentially related disadvantages:(13)$$h=W_{h2}\alpha \left(LN\left(W_{h1}v+b_{h1}\right)\right)+b_{h2}$$

Afterwards, it can be upgraded to the HL h with input gate $i_{g}$:(14)$$h_{r}=h*i_{g}$$
whereas $h_{r}$ implies the upgraded HL.

With $h_{r}$ and $f_{g}$, LSTM cells are calculated across the order:(15)$$c^{t}=c^{t-1}*f_{g}^{t}+h_{r}^{t}\;$$

Equation (15) keeps the step-by-step recurrence upgrade of the LSTM cell and could not be parallelized across the order, and then it only comprises element-wise multiplication–addition functions that are lightweight and, related to linear transformation, could be calculated very quickly on modern hardware.

Different from the original LSTM that calculates the output gate $o_{g}$ dependent upon the concatenated vector $v^{t}$, it calculates the output gate with the recently created cell state c and the input to LSTM, as c is expected to have superior quality to BoW representations.
(16)$$o_{g}=\sigma \left(LN\left(W_{o}i|c+b_{0}\right)\right)\;\;\;$$

Lastly, it executes the output gate to cells and attains the resultant HPLSTM layer.
(17)$$o=c*0_{g}\;$$

Either Equation (16) (comprising the linear transformation to compute the output gate) or Equation (17) is also effectually parallelized in the order.

### 3.4. Parameter Tuning Using the ESMO Algorithm

In this work, the ESMO algorithm helped to properly tune the hyperparameters related to the HP-LSTM model, i.e., learning rate. Shimin Li et al. [25] proposed the SMO algorithm, which is inspired by the diffusion and behaviour conduct of slime Mold in nature. The different steps and phases of SMO comprise approaching food, oscillation, and wrapping food. The mathematical expression of the SMO algorithm is given as follows:

Approach food: in this stage, the odour of food stimulates slime mould (SM) for searching, which leads to massive oscillation and position upgrading and is mathematically given as follows:(18)$$\vec{X\left(t+1\right)}=\left\{\begin{array}{ll} \vec{X_{b}\left(t\right)}+v_{b}\left[W.\vec{X_{A}\left(t\right)}-\vec{X_{B}\left(t\right)}\right], & r<p;\\ \vec{v_{c}}\left[\vec{X\left(t\right)}\right] & r\geq p \end{array}\right.\;\;$$

In Equation (18), $\vec{vc}$ linearly declines from one to zero. $\left[-a,\;a\right]$ defines the magnitude of $\vec{v_{b}}$. ‘*t*’ denotes the current iteration, ‘*W*’ shows the index of weight, $\vec{X_{b}}$ represents the maximal concentration of odour, and $\vec{X_{A}}$ and $\vec{X_{B}}$ denote the two random SMs. ‘*p*’ is evaluated by means of $=\;\tan h\left(\left|q\left(i\right)-F_{best}\right|\right)$. Furthermore, the weight ‘*W*’ is evaluated as follows:(19)$$\vec{W{\left(i\right)}_{smellindex}}=\left\{\begin{array}{l} 1+r\log\;\left(\frac{F_{cbest}-q\left(i\right)}{F_{cbest}-F_{cworst}}+1\right);\;Condition\\ 1-r\log\;\left(\frac{F_{cbest}-q\left(i\right)}{F_{cbest}-F_{cworst}}+1\right);\;Other \end{array}\right.$$

In Equation (19), the condition specifies that $S\left(i\right)$ orders the first half of the population, r signifies the random value in the range of [0,1], bF represents optimum fitness obtained in the existing iterative method, and wF signifies the worst fitness value [26].

Wrap food: the wrap food process in SMO to update the position of SMO is attained as follows:(20)$$\vec{X}=\left\{\begin{array}{l} rand.\left(U_{B}-L_{B}\right)+L_{B},rand<z\\ \vec{X_{b}\left(t\right)}+\vec{v_{b}}.\left(W.\vec{X_{t}^{A}}-\vec{X_{t}^{B}}\right),r<p\\ \vec{v_{c}}.\vec{X_{t}},r\geq p \end{array}\right\}\;\;$$
where LB and UB symbolize the lower and upper limitations nge; rand and r represent the random integer $\left[0,\;1\right]$.

Grabble food: Here, the value of $\vec{\nu b}$ randomly oscillates amongst $\left[-a,\;a\right]$ and progressively approaches 0 as the iteration increases. The values of $\vec{vc}$ oscillate amongst [−1, 1] and eventually tend to zero.

The ESMO technique was designed based on the elite opposition-based learning (EOBL) system, which is an effective and stable system that enhances population variation, broadens the searching area, avoids premature convergence, and strengthens global searching [27]. The searching system employs the possible or reversed solution for assessing the fitness value of prey, afterward ordering the optimum individual for completing the iteration. Considering that the searching agent with the optimum fitness value was regarded as an elite individual, the elite individual was calculated as $x_{e}=\left(x_{e,1},\;x_{e,2},\ldots ,x_{\;e,\;D}\right)$, the possible solution was calculated as $x_{i}=\left(x_{i,1},\;x_{i,2},\ldots ,xi,\;D_{\;}\right)$, and the reverse solution was calculated as $x_{j}=\left(x_{i,1},\;x_{i,2},\ldots ,xi,\;D\right)$. The equation was calculated as:(21)$$x_{i,j}=k\cdot \left(da_{j}+db_{j}\right)-x_{e,j},\;i=1,2,\ldots ,n;j=1,2,\ldots ,D\;$$
whereas n indicates the population size, D refers to the dimensional problem, k implies the arbitrary value that $k\in \left(O,1\right)$, and $da_{j}$ and $db_{j}$ signify the dynamic limits of *j*th decision variable, respectively; this can be calculated as:(22)$$da_{j}=\;\min\;\left(x_{i,j}\right),\;\;db_{j}=\;\max\;\left(x_{i,j}\right)\;\;$$

The dynamic restriction saves an optimum solution and modifies the searching area of the inverse solution.

The searching agent $x_{i,j}$ was calculated as:(23)$$x_{i,j}=rand\left(da_{j},db_{j}\right),\;\;if\;x_{i,j}\langle da_{j}orx_{i,j}\rangle db_{j}$$

## 4. Performance Evaluation

In this section, the resource allocation performance of the ESMOML-RAA model is investigated. To evaluate the performance of the proposed resource allocation scheme, we used Python and TensorFlow for simulation experiments and analysis. The hardware set includes a processor of an i5-4590S CPU@ 3.00 GHz, 1 TB HDD, and 8 GB RAM. The number of base stations (BSs) was 15 with 25 users. Table 2 highlights the system performance assessment of the ESMOML-RAA model with varying numbers of BSs.

Figure 4 represents the system throughput (ST) inspection of the ESMOML-RAA technique with several BSs. The results implied that the ESMOML-RAA method obtained an improved ST with UAVs. For instance, with three BSs, the ESMOML-RAA system with UAVs obtained a higher ST of 117 Mbps. Meanwhile, with seven BSs, the ESMOML-RAA technique with UAVs attained a superior ST of 293 Mbps. Moreover, with 11 BSs, the ESMOML-RAA method with UAVs achieved a higher ST of 617 Mbps.

Figure 5 signifies the energy consumption (ECON) inspection of the ESMOML-RAA technique with numerous BSs; the outcome shows that the ESMOML-RAA system achieved enhanced ECON with UAVs. For example, with three BSs, the ESMOML-RAA method with UAVs attained a higher ECON of 150 Mbps. Meanwhile, with seven BSs, the ESMOML-RAA algorithm with UAVs achieved a greater ECON of 357 Mbps. Furthermore, with 11 BS, the ESMOML-RAA approach with UAVs acquired a maximum ECON of 537 Mbps.

In Figure 6, a comparative system energy efficiency (EE) analysis of the ESMOML-RAA method with other models such as deep Q-Network (DQN), Q-learning, random, and maximum [8] is provided. The experimental outcomes state that the ESMOML-RAA technique reached higher EE values than the other ones. For example, with 10 users, the ESMOML-RAA method attained an improved EE of 147,715,335 bit/j, while the DQN, Q-learning, random, and maximum models obtained a reduced EE of 141,671,107 bit/j, 117,116,431 bit/j, 94,450,577 bit/j, and 84,250,942 bit/j, respectively. Simultaneously, with 25 users, the ESMOML-RAA method achieved a better EE of 26,453,013 bit/j, while the DQN, Q-learning, random, and maximum models acquired a decreased EE of 23,808,663 bit/j, 18,519,963 bit/j, 17,764,435 bit/j, and 15,875,614 bit/j, respectively.

Figure 7 provides a comparative EE examination of the ESMOML-RAA model with other models. The experimental results specify that the ESMOML-RAA technique obtained greater EE values than the others. For example, with three BSs, the ESMOML-RAA method attained an improved EE of 39,213,816 bit/j, while the DQN, Q-learning, random, and maximum models reached decreased EEs of 36,459,435 bit/j, 33,882,755 bit/j, 32,016,884 bit/j, and 28,995,950 bit/j, respectively. Simultaneously, with eight BSs, the ESMOML-RAA technique achieved an enhanced EE of 56,362,059 bit/j, whereas the DQN, Q-learning, random, and maximum approaches obtained reduced EEs of 54,762,741 bit/j, 50,497,893 bit/j, 46,499,598 bit/j, and 42,856,707 bit/j, respectively.

Figure 8 shows an inspection of the overall computation time (CT) of the ESMOML-RAA technique. The obtained values imply that the ESMOML-RAA method attained reduced values of CT under all BSs. For example, with three BSs, the ESMOML-RAA approach provided a minimal CT of 184 s, while the DQN, Q-learning, random, and maximum models reached maximum CTs of 223 s, 261 s, 320 s, and 372 s, respectively. Also, with 15 BSs, the ESMOML-RAA algorithm provided the lowest CT of 393 s, while the DQN, Q-learning, random, and maximum models reached the highest CTs of 455 s, 460 s, 485 s, and 511 s, respectively.

Figure 9 shows an examination of the overall packet loss ratio (PLR) of the ESMOML-RAA model. The attained values show that the ESMOML-RAA model reached decreased values of PLR under all BSs. For example, with three BSs, the ESMOML-RAA model provided a minimal PLR of 30.02%, while the DQN, Q-learning, random, and maximum models attained maximal PLRs of 36.94%, 52.32%, 65.39%, and 75.90%, respectively. Also, with 15 BSs, the ESMOML-RAA model reached the lowest PLR of 9%, whereas the DQN, Q-learning, random, and maximum models provided the highest PLRs of 18.48%, 27.45%, 32.58%, and 38.99%, respectively.

From the detailed results, it is apparent that the ESMOML-RAA model accomplished effectual resource allocation performance.

## 5. Conclusions

In this study, we employed the ESMOML-RAA technique for resource assignment in UAV-enabled wireless networks. The presented ESMOML-RAA technique attained energy-effective and computationally effective decisions proficiently. At the same time, the ESMOML-RAA technique considered the UAV as a learning agent with the formation of resource assignment decisions as actions and designed a reward function with the intention of minimizing of weighted resource consumption. The presented ESMOML-RAA technique employed the HP-LSTM model with the ESMO algorithm as a hyperparameter optimizer for resource allocation. Using the ESMO algorithm helped to properly tune the hyperparameters related to the HP-LSTM module. The performance validation of the ESMOML-RAA technique was tested using a series of simulations. This comparison study reports the enhanced performance of the ESMOML-RAA technique over other ML models.

In future, new resource allocation approaches can be developed to dynamically adapt to varying network conditions in real time such as UAV mobility, varying network traffic, and environmental changes to make proactive decisions and optimize resource allocation accordingly. In addition, the integration of edge computing and federated learning can be investigated in resource allocation for UAV networks.

## Figures and Tables

**Figure 1 sensors-23-07083-f001:**
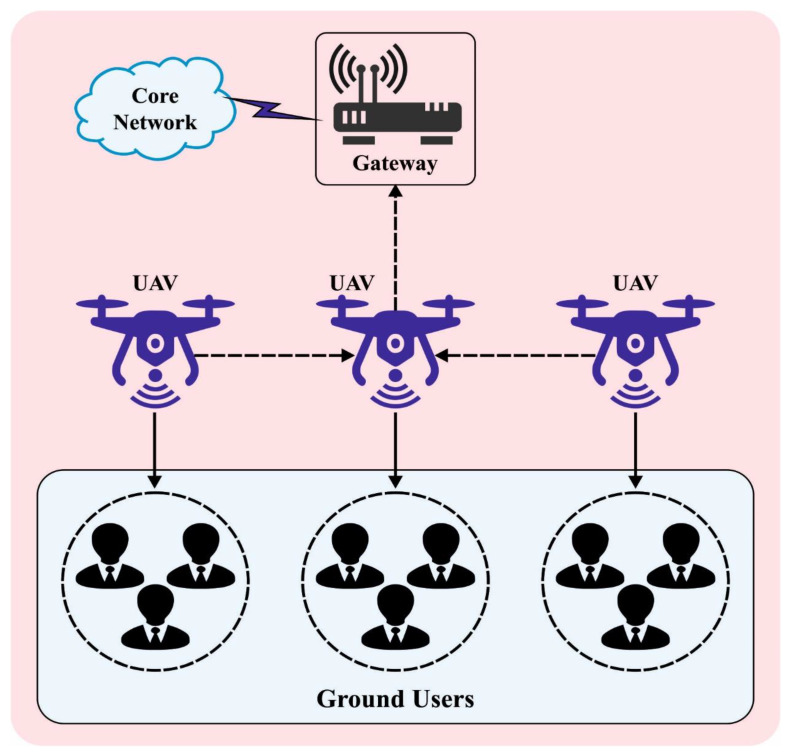
Overview of UAV-enabled WSN.

**Figure 2 sensors-23-07083-f002:**
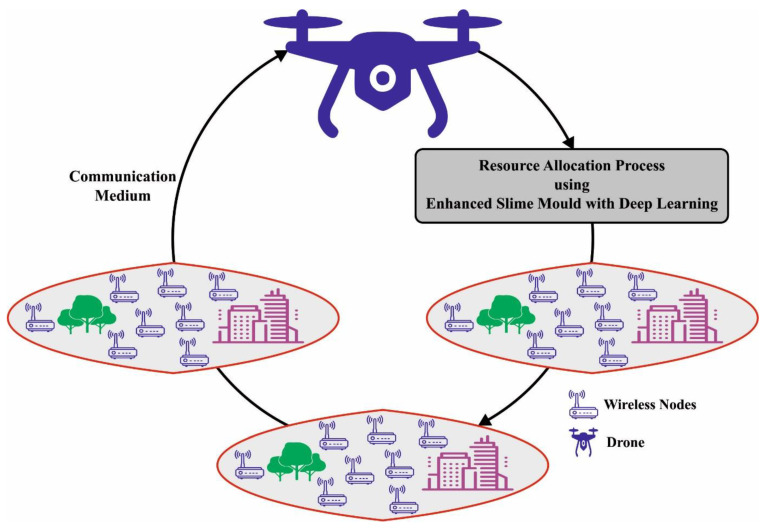
Working process of ESMOML-RAA technique.

**Figure 3 sensors-23-07083-f003:**
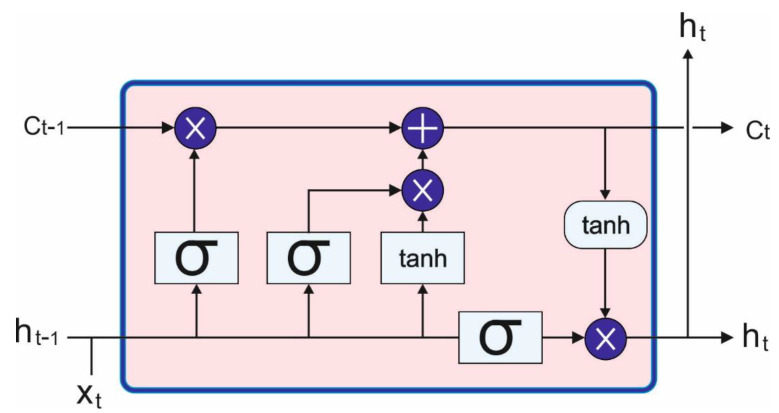
Architecture of LSTM.

**Figure 4 sensors-23-07083-f004:**
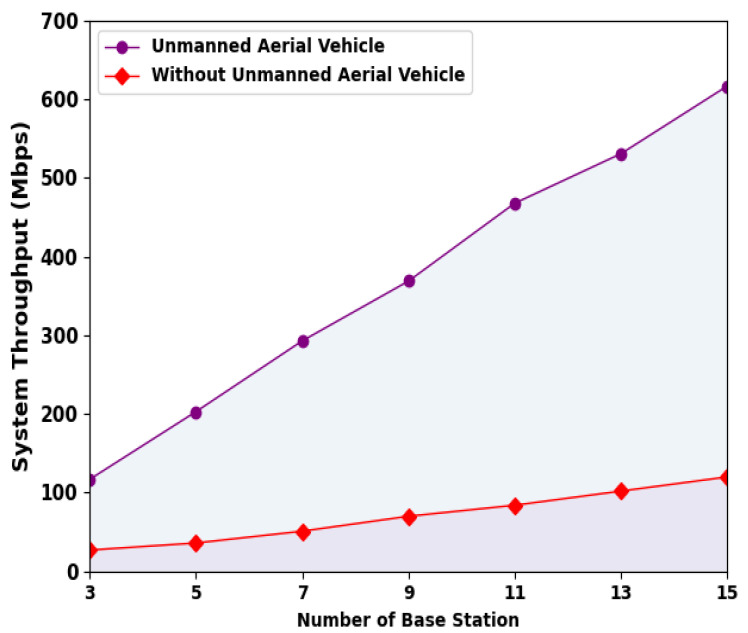
ST analysis of ESMOML-RAA approach with varying numbers of BSs.

**Figure 5 sensors-23-07083-f005:**
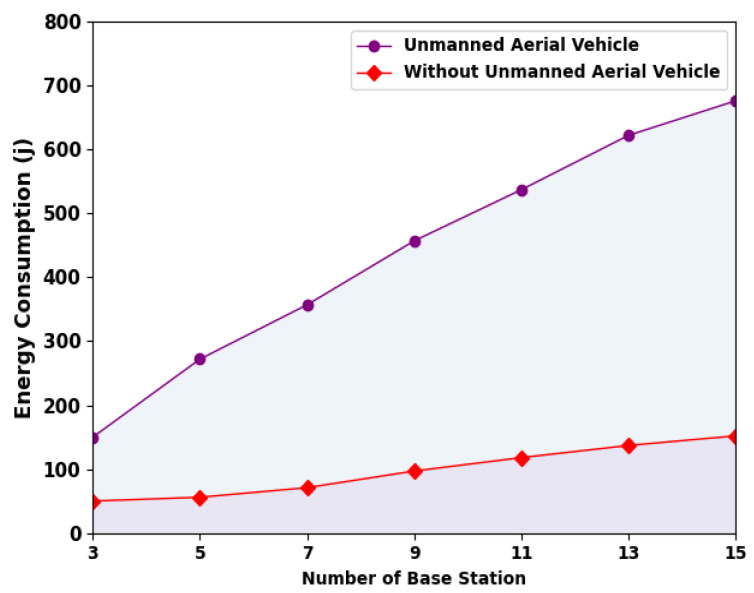
ECON analysis of ESMOML-RAA approach with varying BS.

**Figure 6 sensors-23-07083-f006:**
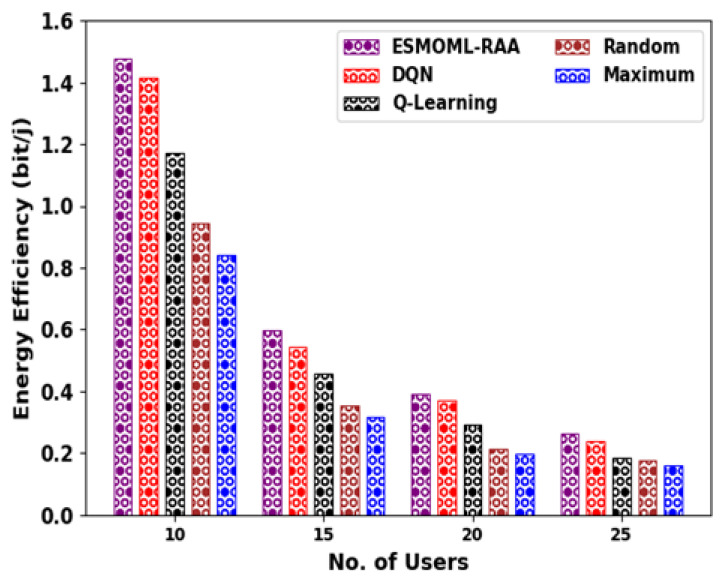
EE analysis of ESMOML-RAA approach with distinct users.

**Figure 7 sensors-23-07083-f007:**
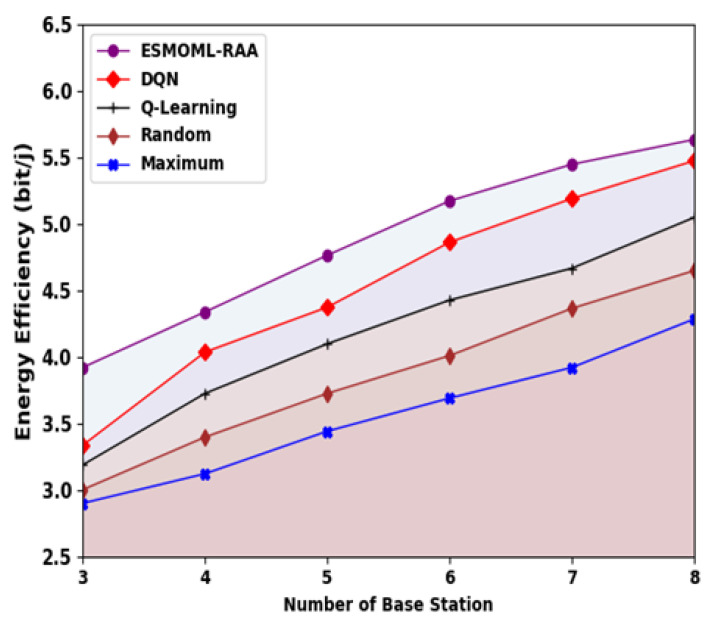
EE analysis of ESMOML-RAA approach with varying numbers of BSs.

**Figure 8 sensors-23-07083-f008:**
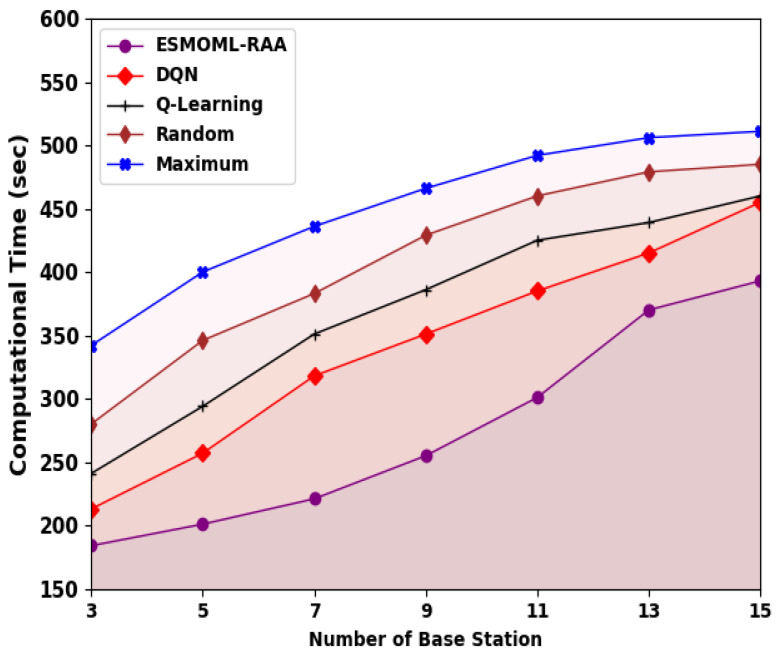
CT analysis of ESMOML-RAA approach with different numbers of BSs.

**Figure 9 sensors-23-07083-f009:**
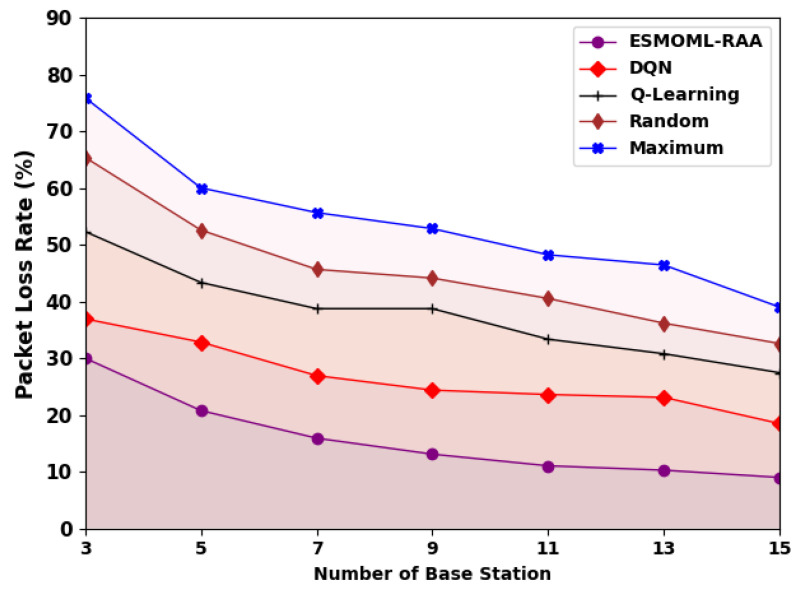
PLR analysis of ESMOML-RAA approach with varying numbers of BSs.

**Table 1 sensors-23-07083-t001:** Notations and descriptions.

Notation	Description
RA	Resource allocation
K	Network function
$T^{\gamma }$	Time interval
i	Input
$d_{0}$	Reference distance
$T^{\gamma }$	Time interval
D	Latency-intensive tasks
B	Bandwidth
$p_{i_{;}k}^{j,s}$	Power consumption

**Table 2 sensors-23-07083-t002:** Result analysis of ESMOML-RAA approach with varying numbers of BSs.

Number of Base Stations	Unmanned Aerial Vehicle		Without Unmanned Aerial Vehicle	
	System Throughput (Mbps)	Energy Consumption (j)	System Throughput (Mbps)	Energy Consumption (j)
3	117	150	27	50
5	203	272	36	56
7	293	357	51	71
9	369	457	70	97
11	468	537	84	118
13	531	622	102	137
15	617	676	120	152

## Data Availability

The data are contained within the article and/or available from the corresponding author upon reasonable request.

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
