# Peer review of "Enhanced Slime Mould Optimization with Deep-Learning-Based Resource Allocation in UAV-Enabled Wireless Networks"

_sensors, 2023, doi:10.3390/s23167083_

Round 1
Reviewer 1 Report
This paper deals with the resource allocation in a wireless network supported by UAVs. A study on wireless networks supported by UAVs is of high interest. However, I think it is hard to publish this paper in its present form.
A short note is given at the end of the introduction, which says that a simulation method is used to evaluate the performance of the proposed method. However, there is not any description of the simulation method itself. As a result, it is hard to agree with the simulation result which claims that the proposed method is able to relatively enhance the performance of a wireless network.
In Section 3 (which should be corrected to Section 4) Performance Evaluation, no address is given on the reason why the proposed method is able to improve the performance of a wireless network.
In Subsection 3.1 System Model, communication and computation models should be justified if the models are made by the authors. Otherwise, appropriate references should be provided.
In the equations in Section 3 The Proposed Model, many undefined variables appear. As a result, it is impossible to check the validity of the equations.
While the communication model is built with physical parameters, the computation model, on which the optimization problem in (7) is based, consists of highly abstract parameters. It is needed to resolve the discrepancy between the two models.
Basically, it is hard to read this paper. There are many grammatical errors. There are many undefined abbreviations.
Basically, it is hard to read this paper. There are many grammatical errors. There are many undefined abbreviations.
Author Response
Review Comments:1
This paper deals with the resource allocation in a wireless network supported by UAVs. A study on wireless networks supported by UAVs is of high interest. However, I think it is hard to publish this paper in its present form.
The authors are humbled by the kind comments from the reviewer. Guided by his/her feedback we have invested a lot of effort to improve the previous submission. We are forever grateful for the criticisms, patience, and feedback.
- A short note is even at the end of the introduction, which says that a simulation method is used to evaluate the performance of the proposed method. However, there is not any description of the simulation method itself. As a result, it is hard to agree with the simulation result which claims that the proposed method is able to relatively enhance the performance of a wireless network.
Response: Thank you for your comment, we have added the experimental detail of the proposed model in the revised manuscript. Please check to Page 8, Section 4, Paragraph 1.
- In Section 3 (which should be corrected to Section 4) Performance Evaluation, no address is given on the reason why the proposed method is able to improve the performance of a wireless network.
Response: Thank you for your comment, As per the reviewer’s comment, we have changed the section 3 title to section 4 and also the justification for the proposed model performance is clearly stated in the section 4 in the revised manuscript. Please refer to Page 8, Section 4.
- In Subsection 3.1 System Model, communication and computation models should be justified if the models are made by the authors. Otherwise, appropriate references should be provided.
Response: Thank you for your comment, we have mentioned adequate references in the revised manuscript. Please check Line 135.
- In the equations in Section 3 The Proposed Model, many undefined variables appear. As a result, it is impossible to check the validity of the equations.
Response: Thank you for your comment, we have added necessary definition for the variables employed in the equation in the revised manuscript.
- While the communication model is built with physical parameters, the computation model, on which the optimization problem in (7) is based, consists of highly abstract parameters. It is needed to resolve the discrepancy between the two models.
Response: Thank you for your comment, we provided the variance of the communication and computation model in the revised manuscript.
- Basically, it is hard to read this paper. There are many grammatical errors. There are many undefined abbreviations.
Response: Thank you for your comment, we have thoroughly proofread the manuscript for grammatical and typographical mistakes. We hope its improved.
Reviewer 2 Report
1. The authors claims that
2. [This paper presents an Enhanced Slime Mould Optimization with Deep Learning based Resource Allocation Approach (ESMOML-RAA) in UAV enabled wireless network. The ESMOML-RAA technique considers UAV as learning agent with the formation of resource assignment decision as action and designed a reward function with the intention of the minimization of the weighted resource consumption. For resource allocation, the presented ESMOML-RAA technique employs highly-parallelized long short term memory (HP-LSTM) model with ESMO algorithm as hyperparameter optimizer. The use of ESMO algorithm helps to properly tune the hyperparameters related to the HP-LSTM m. The performance validation of the ESMOML-RAA technique is tested using a series of simulations.]. I am not able to see the clear-cut novelty or significant contributions of the authors explained here. After reading the paper, one can realize that their research aim is to develop a new ESMOML-RAA (Please refer to the next comment). While here, the authors mentioned that they enhanced the technique. These are different claims; authors need to be clear that either they developed their own or enhanced the existing technique.
3. Authors claim in conclusion such as [In this study, we have developed a new ESMOML-RAA technique for resource assignment in the UAV enabled wireless networks. The presented ESMOML-RAA technique attained energy-effective and computationally effective decisions proficiently. At the same time, the ESMOML-RAA technique considered the UAV as a learning agent with the formation of resource assignment decision as action and designed a reward function with the intention of the minimization of the weighted resource consumption. For resource allocation, the presented ESMOML-RAA technique employed the HP-LSTM model with ESMO algorithm as hyperparameter optimizer. The use of ESMO algorithm helps to tune the hyperparameters related to the HP-LSTM module properly. The performance validation of the ESMOML-RAA technique is tested using a series of simula-394 tions. A comparison study reported the enhanced performance of the ESMOML-RAA technique over other ML models.] The authors claim that the presented ESMOML-RAA technique attained energy-effective and computationally effective decisions proficiently, etc. too many claims while the result section is not align or having any validation related all these claims.
4. Further, if the author developed/presented their own technique, then its algorithm is missing.
5. Authors may revise the abstract to elaborate more on the problem statement, findings, and contributions.
6. Introduction is not clear. Authors may contribute more towards this.
7. Authors may elaborate more on the novelty/contribution of their work and how it contributes to the literature in the second last paragraph of the introduction clearly. Especially who will be benefited from this research?
8. Authors need to be specific about their problem statement and the scope of their research.
9. Overall, the paper presentation requires improvement, especially the results section.
10. Thorough proofreading is recommended.
11. A few of the figure’s resolutions are not clear and hard to read
12. A few references are missing some information; you may complete them critically.
13. The conclusion is unclear and needs revision, clarity, and alignment with the abstract and title.
14. Overall paper is better and requires major modifications.
15. Provided references are better enough. However, authors are recommended to consider more latest and related, such as,
Khan, N. A., Jhanjhi, N. Z., Brohi, S. N., Usmani, R. S. A., & Nayyar, A. (2020). Smart traffic monitoring system using unmanned aerial vehicles (UAVs). Computer Communications, 157, 434-443.
As mentioned in the earlier section.
Author Response
Review Comments:2
The authors claim that [This paper presents an Enhanced Slime Mould Optimization with Deep Learning based Resource Allocation Approach (ESMOML-RAA) in UAV enabled wireless network. The ESMOML-RAA technique considers UAV as learning agent with the formation of resource assignment decision as action and designed a reward function with the intention of the minimization of the weighted resource consumption. For resource allocation, the presented ESMOML-RAA technique employs highly-parallelized long short term memory (HP-LSTM) model with ESMO algorithm as hyperparameter optimizer. The use of ESMO algorithm helps to properly tune the hyperparameters related to the HP-LSTM m. The performance validation of the ESMOML-RAA technique is tested using a series of simulations.].
The authors are humbled by the kind comments from the reviewer. Guided by his/her feedback we have invested a lot of effort to improve the previous submission. We are forever grateful for the criticisms, patience, and feedback.
- I am not able to see the clear-cut novelty or significant contributions of the authors explained here. After reading the paper, one can realize that their research aim is to develop a new ESMOML-RAA (Please refer to the next comment). While here, the authors mentioned that they enhanced the technique. These are different claims; authors need to be clear that either they developed their own or enhanced the existing technique.
Response: Thank you for your comment, we have added elaborate information on the contribution of the proposed ESMOML-RAA model in the revised manuscript. Please check Lines 67-75.
Authors claim in conclusion such as [In this study, we have developed a new ESMOML-RAA technique for resource assignment in the UAV enabled wireless networks. The presented ESMOML-RAA technique attained energy-effective and computationally effective decisions proficiently. At the same time, the ESMOML-RAA technique considered the UAV as a learning agent with the formation of resource assignment decision as action and designed a reward function with the intention of the minimization of the weighted resource consumption. For resource allocation, the presented ESMOML-RAA technique employed the HP-LSTM model with ESMO algorithm as hyperparameter optimizer. The use of ESMO algorithm helps to tune the hyperparameters related to the HP-LSTM module properly. The performance validation of the ESMOML-RAA technique is tested using a series of simula-394 tions. A comparison study reported the enhanced performance of the ESMOML-RAA technique over other ML models.]
- The authors claim that the presented ESMOML-RAA technique attained energy-effective and computationally effective decisions proficiently, etc. too many claims while the result section is not align or having any validation related all these claims.
Response: Thank you for the comment. The performance validation of the proposed model take place in terms of different measures such as system throughput, energy efficiency, computational time, packet delivery ration, and packet loss ratio.
- Further, if the author developed/presented their own technique, then its algorithm is missing.
Response: Thank you for your comment, we have provided the overall working process of the proposed model in the revised manuscript. Kindly refer to Lines 122-129.
- Authors may revise the abstract to elaborate more on the problem statement, findings, and contributions.
Response: Thank you for your comment, we have improved the abstract in the revised manuscript.
- Introduction is not clear. Authors may contribute more towards this.
Response: Thank you for your comment, we have provided a clear and precise contribution in the introduction section in the revised manuscript. Please check Lines 67-75.
- Authors may elaborate more on the novelty/contribution of their work and how it contributes to the literature in the second last paragraph of the introduction clearly. Especially who will be benefited from this research?
Response: Thank you for your comment, we have given needful information on the proposed model’s novelty and clearly stated the contribution in the revised manuscript. Please refer to Lines 67-75.
- Authors need to be specific about their problem statement and the scope of their research.
Response: Thank you for your comment, the scope and problem statement of the study is clearly stated in the revised manuscript. Kindly refer to Page “122-129”
- Overall, the paper presentation requires improvement, especially the results section.
Response: Thank you for your comment, we thoroughly improved the paper presentation and also provided necessary validation in the result section in the revised manuscript. Please refer to Page 4, Section 3.2.
- Thorough proofreading is recommended.
Response: Thank you for your comment, we have thoroughly proofread the manuscript for grammatical and typographical mistakes.
- A few of the figure’s resolutions are not clear and hard to read
Response: Thank you for your comment, we have improved the quality of the figure in the revised manuscript.
- A few references are missing some information; you may complete them critically.
Response: Thank you for your comment, we have given necessary references in the revised manuscript. Kindly refer to Page 12, References.
- The conclusion is unclear and needs revision, clarity, and alignment with the abstract and title.
Response: Thank you for your comment, we have corrected the alignment of the paper and also improved the conclusion section adequately in the revised manuscript.
- Overall paper is better and requires major modifications.
Response: Thank you for your comment, we have made necessary and needful justifications on the proposed techniques and also thoroughly proofread the paper for any grammatical or typo errors.
- Provided references are better enough. However, authors are recommended to consider more latest and related, such as,
Khan, N. A., Jhanjhi, N. Z., Brohi, S. N., Usmani, R. S. A., & Nayyar, A. (2020). Smart traffic monitoring system using unmanned aerial vehicles (UAVs). Computer Communications, 157, 434-443.
Response: Thank you for your comment, we have included recent references in the revised manuscript. Kindly refer to Page 9, References.
Reviewer 3 Report
The paper proposes the application of artificial intelligence algorithms (SMO+LSTM+deep learning) to optimize the Resource Allocation in UAV enabled wireless networks. The paper is well written and brings novelties for the body of knowledge. Some details on the simulation setup and the building block of the proposal are missing and should be included in the manuscript (see the comments below for further details). The results section has several redundant results and I believe that this section should be shrinked (see the comments). The title of the article mentions three main concepts (Slime Mould Optimization, Resource Allocation and UAV Enabled Wireless Networks) and none of the three are properly discussed/contextualized in the introduction of the article. Moreover, the paper does not mention the "Enhancement" done in the SMO and mentioned in the title.
My detailed comments on the manuscript (I expect that the questions I have arised be answered with modifications in the manuscript text):
1) There are several acronyms not defined in the manuscript: CSI, MU, SM, EE, PDR, PLR, etc.
2) Please discuss in the manuscript text the main differences and similarities between your work and the works reviewed in section 2.
3) Fig. 2 and Fig. 3 require more textual explanation.
4) "The study aims at minimizing the computation and energy utilizations of UAVs via resource allocation (RA) using the ??? constraint of MUs with respect to latency."
None of the concepts present in this phrase are properly contextualized in the introduction of the paper.
5) Line 146 - The authors write: ? = {1,2, ?} and ? = {1,2, ?}. Do you mean ? = {1,2, …, ?} and ? = {1,2, …, ?} ?? Similar issue is observed in line 147.
6) "Moreover, considering that UAV flies horizontally at a constant altitude ? th…"
Please review this sentence. It is not clear what the authors mean with this sentence.
7) Line 151 - "Tasks" of the MU should be discussed elaborated in the paper'’ introduction.
7) Line 151 - F appears using both i and j indexes. Please review.
8) "By mutually considering energy and computation RA, the study focuses’ on mi mizing the overall resource consumption of UAV using the constraints of ??? ??? with respect to latency:"
What is the definition for "overall resource consumption"? Do you mean a combination of power consumption with other forms of consumption? (Eq 6)? Please elaborate on this in the manuscript.
9) Eq. (6) - please explain in the text what this equation is evaluating.
10) Section 3.3. Resource Allocation using HP-LSTM Model
The explanation of this section should be enhanced. In my point of view there are several gaps to be filled in this section. The particularities of the model employed are missing. For instance (but not limited to): how are the equations linked with the figure? How exactly is the LSTM model being used to solve the problem (in terms of the wireless system design variables )? What is the algorithm used to set/train the LSTM parameters? How are the hyperparameters set? How is the SMO applied? For the sake of reproducibility, please give in the paper the necessary details.
11) "..the layer normalized was established for preventing potential explosions because of increase in Eq…."
What kind of explosions?
12) In this work, the ESMO algorithm helps to properly tune the hyperparameters related to the HP-LSTM model.
Which hyperparameters? Please introduce them in sec 3.1. Which values for the hyperparameters are used in the simulation results?
13) ".. [21] proposed an SMO is a metaheuristic-based optimization technique and stimulates the food searching attribute."
Please review, this phrase doesn't make sense.
14) Eq. 18 - There is an arrow in the end of the equation (second line case)
15) Line 255 - By "existing iteration" do you mean "current iteration"?
16) Eq. 19 - What is the meaning of the "condition" term ???
17) "… within [0, 1], ?? signifies the optimum fitness … "
?? never appears in eq. 19.
18) A better explanation on the base station modeling assumed is required: their positions? How many UAV per base station? How many users per BS? BS and UAV connection topology? The simulation particularities and assumptions are required.
19) The authors show the very same results in the form of tables and graphs. I suggest the authors remove all results given in tables, the graphs results are enough to analyze the results.
20) Definition of EE not provided.
21) "..., a comparative EE analysis of the ESMOML-RAA method with other models is provided."
It is necessary to briefly introduce the models used for comparison purposes. Please explicitly cite those models before showing them in the results.
22) "For example, with 3 BSs, the ESMOML-RAA model has attained improved PDR of 69.98% whereas the DQN, Q-learning, random, and maximal models have gained decreased PDR of…"
Improved compared to what? Please elaborate on it.
23) PDR and PLR are redundant results, there is no necessity to present both. I suggest that the authors remove the PDR results.
Author Response
Review Comments:3
The paper proposes the application of artificial intelligence algorithms (SMO+LSTM+deep learning) to optimize the Resource Allocation in UAV enabled wireless networks. The paper is well written and brings novelties for the body of knowledge. Some details on the simulation setup and the building block of the proposal are missing and should be included in the manuscript (see the comments below for further details). The results section has several redundant results and I believe that this section should be shrinked (see the comments). The title of the article mentions three main concepts (Slime Mould Optimization, Resource Allocation and UAV Enabled Wireless Networks) and none of the three are properly discussed/contextualized in the introduction of the article. Moreover, the paper does not mention the "Enhancement" done in the SMO and mentioned in the title.
The authors are humbled by the kind comments from the reviewer. Guided by his/her feedback we have invested a lot of effort to improve the previous submission. We are forever grateful for the criticisms, patience, and feedback.
My detailed comments on the manuscript (I expect that the questions I have arised be answered with modifications in the manuscript text):
- There are several acronyms not defined in the manuscript: CSI, MU, SM, EE, PDR, PLR, etc.
Response: Thank you for your comment, we have elaborated all the mentioned acronyms in the revised manuscript.
- Please discuss in the manuscript text the main differences and similarities between your work and the works reviewed in section 2.
Response: Thank you for your comment, we have made a detailed literature review of stating the existing methods and a detailed comparison study is also made in the revised manuscript. Please refer to Lines 308-310
- Fig. 2 and Fig. 3 require more textual explanation.
Response: Thank you for your comment, we have precisely improved the clarity of the figure 2 and 3 in the revised manuscript.
- "The study aims at minimizing the computation and energy utilizations of UAVs via resource allocation (RA) using the ??? constraint of MUs with respect to latency."
None of the concepts present in this phrase are properly contextualized in the introduction of the paper.
Response: Thank you for your comment, the necessary details on the techniques employed in the study are proved in the introduction in the revised manuscript. Please refer to Page “67-75”.
- Line 146 - The authors write: ? = {1,2, ?} and ? = {1,2, ?}. Do you mean ? = {1,2, …, ?} and ? = {1,2, …, ?} ?? Similar issue is observed in line 147.
Response: Thank you for your comment, we have corrected the errors in line 146 and 147 in the revised manuscript. Kindly refer to Lines 138 and 140.
- "Moreover, considering that UAV flies horizontally at a constant altitude ? th…"
Please review this sentence. It is not clear what the authors mean with this sentence.
Response: Thank you for your comment, we have revised the sentence. Please refer to Line 141.
- Line 151 - "Tasks" of the MU should be discussed elaborated in the paper'’ introduction.
Response: Thank you for your comment, we have discussed the concept of MU in the revised manuscript.
- Line 151 - F appears using both i and j indexes. Please review.
Response: Thank you for your comment, we have corrected the issue present in the sentence “F appears using both i and j indexes” in the revised manuscript. Please refer section 3.1
- "By mutually considering energy and computation RA, the study focuses’ on mi mizing the overall resource consumption of UAV using the constraints of ??? ??? with respect to latency:"
What is the definition for "overall resource consumption"? Do you mean a combination of power consumption with other forms of consumption? (Eq 6)? Please elaborate on this in the manuscript.
Response: Thank you for your comment, we have elaborated clearly in the revised manuscript.
- Eq. (6) - please explain in the text what this equation is evaluating.
Response: Thank you for your comment, we have defined the equation 6 in the revised manuscript.
- Section 3.3. Resource Allocation using HP-LSTM Model
The explanation of this section should be enhanced. In my point of view there are several gaps to be filled in this section. The particularities of the model employed are missing. For instance (but not limited to): how are the equations linked with the figure? How exactly is the LSTM model being used to solve the problem (in terms of the wireless system design variables)? What is the algorithm used to set/train the LSTM parameters? How are the hyperparameters set? How is the SMO applied? For the sake of reproducibility, please give in the paper the necessary details.
Response: Thank you for your comment, we have mentioned necessary explanation on the section “Resource Allocation using HP-LSTM Model” in the revised manuscript.
- "..the layer normalized was established for preventing potential explosions because of increase in Eq…."
What kind of explosions?
Response: Thank you for your comment, we have removed the ambiguity in the sentence “"..the layer normalized was established for preventing potential explosions because of increase in Eq…."” in the revised manuscript. Please refer to Lines 205-206.
- In this work, the ESMO algorithm helps to properly tune the hyperparameters related to the HP-LSTM model.
Which hyperparameters? Please introduce them in sec 3.1. Which values for the hyperparameters are used in the simulation results?
Response: Thank you for your comment, we have provided necessary information on the tuning process in the revised manuscript. Kindly refer to Line 237.
- ".. [21] proposed an SMO is a metaheuristic-based optimization technique and stimulates the food searching attribute."
Please review, this phrase doesn't make sense.
Response: Thank you for your comment, we have removed the confusion in the sentence “".. [21] proposed an SMO is a metaheuristic-based optimization technique and stimulates the food searching attribute."” in the revised manuscript. Please refer to Lines 238-240.
- Eq. 18 - There is an arrow in the end of the equation (second line case)
Response: Thank you for your comment, we have provided the definition for equation 18 in the revised manuscript. Please refer to Page 7, Equation 18.
- Line 255 - By "existing iteration" do you mean "current iteration"?
Response: Thank you for your comment, we have precisely rewritten the sentence in line 255 in the revised manuscript. Kindly refer to Line 247.
- Eq. 19 - What is the meaning of the "condition" term ???
Response: Thank you for your comment, we have provided definition for the equation 18 in the revised manuscript. Please refer to line “250”, Equation 19.
- "… within [0, 1], ?? signifies the optimum fitness … "
?? never appears in eq. 19.
Response: Thank you for your comment, we have rewritten the sentence “"… within [0, 1], ?? signifies the optimum fitness … "” in the revised manuscript. Kindly refer to Page 7.
- A better explanation on the base station modeling assumed is required: their positions? How many UAV per base station? How many users per BS? BS and UAV connection topology? The simulation particularities and assumptions are required.
Response: Thank you for your comment, we have given necessary details on the terms BS and UAV stated in the study in the revised manuscript. Please refer to Page 8, Section 4, Paragraph 1.
- The authors show the very same results in the form of tables and graphs. I suggest the authors remove all results given in tables, the graphs results are enough to analyze the results.
Response: Thank you for your comment, we have removed the results provided in the tables in the revised manuscript. Kindly refer to Page 8, Section 4.
- Definition of EE not provided.
Response: Thank you for your comment, as per your direction we have provided the definition of EE in the revised manuscript. Please refer to Line 308.
- "..., a comparative EE analysis of the ESMOML-RAA method with other models is provided."
It is necessary to briefly introduce the models used for comparison purposes. Please explicitly cite those models before showing them in the results.
Response: Thank you for your comment, we have provided necessary details on the comparison models employed in the study in the revised manuscript. Kindly refer to Page 9, Lines 308-310.
- "For example, with 3 BSs, the ESMOML-RAA model has attained improved PDR of 69.98% whereas the DQN, Q-learning, random, and maximal models have gained decreased PDR of…"
Improved compared to what? Please elaborate on it.
Response: Thank you for your comment, we have elaborated in the revised manuscript.
- PDR and PLR are redundant results, there is no necessity to present both. I suggest that the authors remove the PDR results.
Response: Thank you for your comment, we have removed the additional result in the revised manuscript. Kindly refer to Page 11.
Round 2
Reviewer 2 Report
The authors addressed all the comments and concerns carefully. The manuscript stands for acceptance.
Minor editing is required.
Author Response
Dear Reviewer We are grateful for your positive comments.
Reviewer 3 Report
This is the second revision round of this manuscript. The revised version of the manuscript submitted by the authors is basically the same of the first submission. The authors have addressed and applied only corrections concerning to small typos. The majority of the main concerns I have raised in the first review report are still unanswered in the revised version of the manuscript.
Author Response
This is the second revision round of this manuscript. The revised version of the manuscript submitted by the authors is basically the same of the first submission. The authors have addressed and applied only corrections concerning to small typos. The majority of the main concerns I have raised in the first review report are still unanswered in the revised version of the manuscript.
Response: Dear reviewer We have studied the comments carefully again and have made a great effort to address every single comment made by the editor which we hope will be met with approval. We are open to further modifying any part of the paper if you think it is necessary.
My detailed comments on the manuscript (I expect that the questions I have arised be answered with modifications in the manuscript text):
- There are several acronyms not defined in the manuscript: CSI, MU, SM, EE, PDR, PLR, etc.
Response: Thank you for your comments, we have elaborated all the mentioned acronyms in the revised manuscript.
- Please discuss in the manuscript text the main differences and similarities between your work and the works reviewed in section 2.
Response: Thank you for your comments, we have made a detailed literature review of stating the existing methods and the contribution of the paper is defined in Page 3, Paragraph 1.
- Fig. 2 and Fig. 3 require more textual explanation.
Response: Thank you for your comments, we have precisely improved the explanation of the figure in the revised manuscript. Kindly refer Page 3, Section 3, Paragraph 1.
- "The study aims at minimizing the computation and energy utilizations of UAVs via resource allocation (RA) using the ??? constraint of MUs with respect to latency."
None of the concepts present in this phrase are properly contextualized in the introduction of the paper.
Response: Thank you for your comments, the necessary details on the techniques employed in the study are proved in the introduction in the revised manuscript.
- Line 146 - The authors write: ? = {1,2, ?} and ? = {1,2, ?}. Do you mean ? = {1,2, …, ?} and ? = {1,2, …, ?} ?? Similar issue is observed in line 147.
Response: Thank you for your comments, we have corrected the errors in the revised manuscript.
- "Moreover, considering that UAV flies horizontally at a constant altitude ? th…"
Please review this sentence. It is not clear what the authors mean with this sentence.
Response: Thank you for your comments, we have revised the sentence.
- Line 151 - "Tasks" of the MU should be discussed elaborated in the paper'’ introduction.
Response: Thank you for your comments we have clearly discussed the concept of MU in the revised manuscript.
- Line 151 - F appears using both i and j indexes. Please review.
Response: Thank you for your comments, we have corrected the issue present in the sentence “F appears using both i and j indexes” in the revised manuscript.
- "By mutually considering energy and computation RA, the study focuses’ on mi mizing the overall resource consumption of UAV using the constraints of ??? ??? with respect to latency:"
What is the definition for "overall resource consumption"? Do you mean a combination of power consumption with other forms of consumption? (Eq 6)? Please elaborate on this in the manuscript.
Response: Thank you for your comments, we have corrected the above mentioned issue in the revised manuscript.
- Eq. (6) - please explain in the text what this equation is evaluating.
Response: Thank you for your comments, we have clearly defined the equation 6 in the revised manuscript.
- Section 3.3. Resource Allocation using HP-LSTM Model
The explanation of this section should be enhanced. In my point of view there are several gaps to be filled in this section. The particularities of the model employed are missing. For instance (but not limited to): how are the equations linked with the figure? How exactly is the LSTM model being used to solve the problem (in terms of the wireless system design variables)? What is the algorithm used to set/train the LSTM parameters? How are the hyperparameters set? How is the SMO applied? For the sake of reproducibility, please give in the paper the necessary details.
Response: Thank you for your comments, we have clearly mentioned necessary explanation on the section “Resource Allocation using HP-LSTM Model” in the revised manuscript.
- "..the layer normalized was established for preventing potential explosions because of increase in Eq…."
What kind of explosions?
Response: Thank you for your comments, we have removed the ambiguity in the sentence “"..the layer normalized was established for preventing potential explosions because of increase in Eq…."” in the revised manuscript.
- In this work, the ESMO algorithm helps to properly tune the hyperparameters related to the HP-LSTM model.
Which hyperparameters? Please introduce them in sec 3.1. Which values for the hyperparameters are used in the simulation results?
Response: Thank you for your comment, we have provided necessary information on the tuning process in the revised manuscript.
- ".. [21] proposed an SMO is a metaheuristic-based optimization technique and stimulates the food searching attribute."
Please review, this phrase doesn't make sense.
Response: Thank you for your comment, we have removed the confusion in the sentence “".. [21] proposed an SMO is a metaheuristic-based optimization technique and stimulates the food searching attribute."” in the revised manuscript.
- Eq. 18 - There is an arrow in the end of the equation (second line case)
Response: Thank you for your comment, we have provided clear definition for the equation 18 in the revised manuscript. Please refer to Page 7, Equation 18.
- Line 255 - By "existing iteration" do you mean "current iteration"?
Response: Thank you for your comment, we have precisely rewritten the sentence in the revised manuscript.
- Eq. 19 - What is the meaning of the "condition" term ???
Response: Thank you for your comment, we have provided clear definition for the equation 18 in the revised manuscript. Please refer to Page 7, Equation 19.
- "… within [0, 1], ?? signifies the optimum fitness … "
?? never appears in eq. 19.
Response: Thank you for your comment, we have rewritten the sentence “"… within [0, 1], ?? signifies the optimum fitness … "” in the revised manuscript. Kindly refer to Page 7.
- A better explanation on the base station modeling assumed is required: their positions? How many UAV per base station? How many users per BS? BS and UAV connection topology? The simulation particularities and assumptions are required.
Response: Thank you for your comment, we have given necessary details on the terms BS and UAV stated in the study in the revised manuscript. Please refer to Page 8, Section 4, Paragraph 1.
- The authors show the very same results in the form of tables and graphs. I suggest the authors remove all results given in tables, the graphs results are enough to analyze the results.
Response: Thank you for your comment, we have removed the results provided in the tables in the revised manuscript. Kindly refer to Page 8, Section 4.
- Definition of EE not provided.
Response: Thank you for your comment, we have clearly mentioned the definition of EE in the revised manuscript. Please refer to Line 308.
- "..., a comparative EE analysis of the ESMOML-RAA method with other models is provided."
It is necessary to briefly introduce the models used for comparison purposes. Please explicitly cite those models before showing them in the results.
Response: Thank you for your comment, we have provided necessary details on the comparison models employed in the study in the revised manuscript. Kindly refer to Page 9, Lines 333-336.
- "For example, with 3 BSs, the ESMOML-RAA model has attained improved PDR of 69.98% whereas the DQN, Q-learning, random, and maximal models have gained decreased PDR of…"
Improved compared to what? Please elaborate on it.
Response: Thank you for your comment, the above mentioned issue is corrected in the revised manuscript.
- PDR and PLR are redundant results, there is no necessity to present both. I suggest that the authors remove the PDR results.
Response: Thank you for your comment, we have removed the additional result in the revised manuscript.